# Comparative analysis of diagnostic performance in mammography: A reader study on the impact of AI assistance

**Marlina Tanty Ramli Hamid*****, Nazimah Ab Mumin***, **Shamsiah Abdul Hamid, Natasha Mohd Ariffin**◯, **Khariah Mat Nor, Ernisha Saib, Nurul Amira Mohamed**

Department of Radiology, Faculty of Medicine, University Teknologi MARA, Sungai Buloh, Selangor, Malaysia

* marlina352@uitm.edu.my (MTRH); Nazimah_mumin@uitm.edu.my (NAM)

## Abstract

### Purpose

This study evaluates the impact of artificial intelligence (AI) assistance on the diagnostic performance of radiologists with varying levels of experience in interpreting mammograms in a Malaysian tertiary referral center, particularly in women with dense breasts.

### Methods

A retrospective study including 434 digital mammograms interpreted by two general radiologists (12 and 6 years of experience) and two trainees (2 years of experience). Diagnostic performance was assessed with and without AI assistance (Lunit INSIGHT MMG), using sensitivity, specificity, positive predictive value (PPV), negative predictive value (NPV), and area under the receiver operating characteristic curve (AUC). Inter-reader agreement was measured using kappa statistics.

### Results

AI assistance significantly improved the diagnostic performance of all reader groups across all metrics (p < 0.05). The senior radiologist consistently achieved the highest sensitivity (86.5% without AI, 88.0% with AI) and specificity (60.5% without AI, 59.2% with AI). The junior radiologist demonstrated the highest PPV (56.9% without AI, 74.6% with AI) and NPV (90.3% without AI, 92.2% with AI). The trainees showed the lowest performance, but AI significantly enhanced their accuracy. AI assistance was particularly beneficial in interpreting mammograms of women with dense breasts.

### Conclusion

AI assistance significantly enhances the diagnostic accuracy and consistency of radiologists in mammogram interpretation, with notable benefits for less experienced

**Data availability statement:** The dataset underlying this study has been made publicly available in Figshare in accordance with PLOS data-sharing requirements. The data can be accessed at DOI: 10.6084/m9.figshare.28529132. This dataset includes de-identified summary statistics relevant to the study while ensuring compliance with institutional policies and patient confidentiality regulations.

**Funding:** This research was supported by the Universiti Teknologi MARA Research Grant Scheme: 600/TNCPI-5/3/DDF (MEDIC) (003/2021), 600/TNCPI-5/3/DDJ(HUITM) (003/2021), and 600/RMC-LESTARI SDG-T5/3. The funders had no role in study design, data collection and analysis, decision to publish, or preparation of the manuscript.

**Competing interests:** The authors have declared that no competing interests exist.

readers. These findings support the integration of AI into clinical practice, particularly in resource-limited settings where access to specialized breast radiologists is constrained.

## Introduction

Breast cancer remains a leading cause of morbidity and mortality among women worldwide, emphasizing the critical need for early detection and accurate diagnosis to improve treatment outcomes and survival rates [1]. Mammography is the primary imaging modality for breast cancer screening, but its diagnostic performance is inherently influenced by the radiologist's experience, interpretative skills, and workload pressures [2]. Given these limitations, artificial intelligence (AI) has emerged as a promising tool in medical imaging, offering the potential to augment radiologists' capabilities, improve diagnostic accuracy, reduce human error, and optimize workflow efficiency [3,4].

Despite the increasing body of evidence supporting AI's role in mammography, its impact on diagnostic performance across different levels of clinical expertise, particularly in high-volume, tertiary referral settings, remains an area requiring further exploration [5,6]. In resource-constrained environments like Malaysia—a middle-income country with a limited number of breast radiologists and the absence of routine second-reader programs for screening mammography—these challenges are even more pronounced. AI could serve as a crucial second reader and decision-support tool, enhancing diagnostic accuracy and efficiency where specialized expertise is scarce [7,8].

The implications of AI in mammography extend beyond Malaysia to other low- and middle-income countries facing similar challenges in breast cancer screening. In many regions with a shortage of trained breast radiologists, AI-driven decision-support systems could help bridge the gap by improving diagnostic consistency, particularly among less experienced readers. Understanding how AI performs in the Malaysian context provides valuable insights that may be generalizable to other countries with comparable healthcare infrastructures, supporting its broader adoption in global breast cancer screening programs

This study, conducted at a tertiary referral center in Malaysia, aims to address this gap by evaluating the diagnostic performance of trainee radiologists and general radiologists in interpreting mammograms, both with and without AI assistance. The integration of AI in mammography offers the potential to improve diagnostic accuracy and potentially democratize expertise by supporting less experienced readers, thereby democratizing expertise in breast imaging. However, concerns remain regarding potential over-reliance on AI, its effect on interpretative skills, and the evolving role of radiologists in breast cancer diagnosis [9,10].

This reader study assesses the impact of AI assistance on the diagnostic performance of readers with varying levels of expertise, providing valuable insights into the practical benefits and limitations of AI in mammography. By examining diagnostic outcomes with AI assistance, this study contributes to the ongoing discourse on

optimizing AI implementation in radiology. It also aims to inform policies and guidelines for AI application in mammography, ensuring that such technologies are utilized to maximize patient care, improve diagnostic accuracy, and enhance healthcare accessibility across diverse clinical settings.

## Methodology

### Study design

This was a retrospective cross-sectional study approved by the Institutional Review Board (Medical Research Ethics ID 2022530–11258).) involving mammography cases from January 2021 to May 2022 done in Hospital Al-Sultan Abdullah UiTM. We excluded cases with post operative changes, personal history of breast cancer, incomplete or missing data, and patients lost from follow-up. Clinical data were extracted from electronic medical records (EMR). Data on patients age, ethnicity, breast density, American College of Radiology-Breast Imaging and Reporting and Data Systems classification (BIRADS) assessment and final histopathological (HPE) diagnosis were recorded.

### Equipment and technique

Mammography examination were performed using a 3D digital mammography system (Selenia Dimension, Hologic, Bedford, Massachusetts, USA). All sets consist of craniocaudal (CC) and mediolateral oblique (MLO) views. The processed image data were sent from the acquisition workstation to the reading workstation.

### AI software & algorithm

Lunit INSIGHT MMG (version 1.1.7.2, Lunit, South Korea) was selected for this study based on its FDA approval and prior validation studies, which demonstrated its effectiveness in enhancing breast cancer detection [11,12]. Previous research has shown that Lunit INSIGHT MMG improves diagnostic performance, particularly in increasing sensitivity and aiding radiologists in detecting subtle malignancies [13–16].

The AI system assigns a malignancy probability score ranging from 0% to 100% to each detected lesion. A threshold of 10% was used in this study, where scores below 10% were deemed clinically insignificant based on prior validation studies [12,13]. This means that findings scoring less than 10% were classified as normal or clinically insignificant findings, with no clinical action required. The system prioritizes suspicious findings with higher malignancy scores, assisting radiologists in their decision-making process.

### Data collection and storage

Data were kept in the hospital picture archiving and communication system (PACS) (Infinitt, South Korea). Data were accessed for research purposes from 1st June 2023–28th February 2024. All image datasets were anonymized and subsequently stored in digital imaging and communication in medicine (DICOM) format in a password-protected external hard disk. Images with incomplete datasets, poor diagnostic quality, and post-operative cases were excluded.

### Image assessment protocol

AI technology was not used in routine clinical practice at the time of image acquisition. The AI-assisted interpretations in this study were conducted solely in a research setting, ensuring that the readers had no prior exposure to AI during routine reporting. All readers underwent training on the application of the AI software prior to the research.

Four readers (2 trainee radiologists (ES – R1) and (NAM – R4) and two general radiologists (NMA-R3 and KMN – R2) with 6 and 12 years' experience, independently reviewed same set of mammography images in a randomized order, in two separate sessions: first without AI assistance and then with AI assistance following a six-week washout period to minimize recall bias. Fig 1 illustrates the study design workflow.

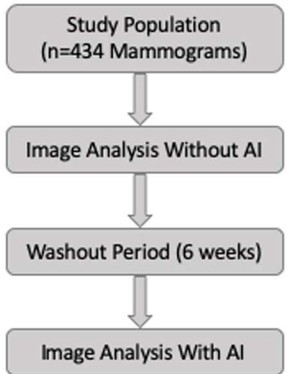

**Fig 1. Study design flowchart illustrating the methodology.** Mammograms were first analysed by radiologists without AI assistance, followed by a six-week washout period before re-evaluation with AI support.

The readers assigned BI-RADS classifications and density assessments based on ACR BI-RADS 5th edition (2013) lexicon which includes density assessment, calcifications, mass, asymmetric density and architectural distortion. All readers are blinded to the radiology report, HPE findings (if performed), and the patients' outcome. A final BIRADS assessment category score were given for each case.

## Statistical analysis

All the data were collected in an Excel spreadsheet, and statistical analysis was performed using IBM SPSS software v.23. A p-value of < 0.05 will be taken as statistically significant. Comparisons between reader groups with and without AI assistance were conducted using Cohen kappa tests (κ). Inter-reader correlation (ICC) coefficient was calculated to ascertain agreement between readers in BI-RADS category assessment and breast density.

Diagnostic performance metrics, including sensitivity, specificity, positive predictive value (PPV), negative predictive value (NPV), and area under the receiver operating characteristic (ROC) curve (AUC), were calculated for each reader group, both with and without AI assistance. The gold standard for final diagnosis was the HPE results if a biopsy was taken. If no biopsy was performed, the cases were followed up for up to 2-year, and considered benign when remains stable.

Sensitivity was when BIRADS 4 or 5 category given by radiologist and HPE results were malignant. Specificity was BIRADS 1,2, or 3 category given by radiologist on imaging and HPE or follow up imaging were benign. Positive predictive value (PPV) were malignant result in HPE and BIRADS category assigned were 4 or 5. Negative predictive value (NPV) were benign result in HPE or on follow-up, and BIRADS category assigned were 1, 2, or 3.

## Ethical approval

This study was approved by the Ethics Committee of University Malaya Medical Centre and Hospital Al-Sultan Abdullah UiTM (Medical Research Ethics ID 2022530–11258) and supported by University Teknologi MARA Research Grant Scheme: 600/TNCPI-5/3/DDF (MEDIC)(003/2021), 600/TNCPI-5/3/DDJ(HUITM)(003/2021) and 600/RMC-LESTARI SDG-T5/3.

## Results

A total of 434 mammography were analysed, consisting of 339 (78.1%) Malays, 71 (16.4%) Chinese, and 24 (5.5%) Indians. The patients' ages ranged from 30 to 80, with a mean age of 52.1 (SD: 0.513).

## HPE findings

Tissue biopsies were performed on 134 (30.8%) breasts whilst 300 (69%) were followed up. All patients had at least 2 years of follow-up after mammogram and/or biopsy. Of the lesions biopsied, 58 (43.2%) were benign and 76 (56.7%) were malignant. 9 (6.7%) B3, premalignant lesions, all of which were papilloma, were included in the benign category for this study.

Table 1 outlines the detailed HPE findings of the lesions. The most common benign lesion were fibroadenoma and fibrocystic change, and the most common malignant lesion was invasive ductal carcinoma. Notably, all reported benign and normal findings showed no interval cancers during a minimum of one and a half year follow-up period.

## Mammogram findings

There were 288 (66.4%) women with dense breast (BIRADS density C and D). Supplementary breast ultrasound was performed in all density C and D patients. Of the 134 biopsied performed, 98 (73.1%) were in dense breast. 45 (81.8%) HPE proven benign and 53 (69.7%) malignant lesions were observed in dense breasts.

Table 2 outlines the lesion presentation on mammogram and HPE. Masses, calcifications and architectural distortion were observed in a higher proportion of malignant compared to benign cases.

There was moderate to substantial agreement in density reading between the readers and AI software with K value 0.549–0.627, p =<0.001.

**Table 1. Histopathology findings of the lesions (n = 134).**

| Benign | n (%)<br>58 (43.2) | Malignant | n (%)<br>76 (56.7) |
|---|---|---|---|
| Fibroadenoma | 10 (17.2) | Invasive ductal carcinoma (IDC) | 68 (89.5) |
| Fibrocystic disease | 10 (17.2) | Ductal carcinoma in situ (DCIS) | 8 (10.5) |
| Adenosis | 7 (12.1) | | |
| Intraductal papilloma | 9 (15.5) | | |
| Usual ductal hyperplasia (UDH) | 4 (6.9) | | |
| Pseudoangiomatous stromal hyperplasia (PASH) | 2 (3.4) | | |
| Benign breast tissue | 9 (15.5) | | |
| Other benign lesions* | 7 (12.1) | | |

*mastopathy, fibrosis, foreign body reaction, lactational changes, mastitis, granuloma, cyst

**Table 2. Histopathology and mammographic lesion presentation overall and in dense breasts.**

| Lesion presentation on mammogram | Benign<br>n = 58 (43.2%) | Malignant<br>n = 76*<br>(56.7%) | Total<br>(n = 134) | Benign<br>n = 43 (44.8%) | Malignant<br>n = 53*<br>(52.2%) | Total<br>(n = 96) |
|---|---|---|---|---|---|---|
| | ALL CASES | | | DENSE BREAST | | |
| Mass | 9 (15.5) | 32 (42.1) | 41 | 4 (9.3) | 15 (28.3) | 19 |
| Calcifications | 14 (24.1) | 26 (34.2) | 40 | 11(25.6) | 20 (37.7) | 31 |
| Architectural distortion | 2 (3.4) | 16 (21.0) | 18 | 1 (2.3) | 10 (18.9) | 11 |
| Asymmetry | 0 | 6 (0.8) | 6 | 0 | 5 (9.4) | 5 |
| Only seen on ultrasound | 33 (56.9) | 0 | 33 | 27 (62.8) | 3 (5.7) | 30 |

*there are 4 cases in malignant group with overlapping findings, which is mass with calcifications

A significant negative correlation were seen between breast density and mass detection on mammograms on Spearman correlation indicating that as breast density increases, mass detection on mammograms tends to decrease, or vice versa. On the other hand, significant positive correlation were seen between breast density and detection of calcifications on mammograms.

## Final BI-RADS assessment category

Intraclass Correlation Coefficient (ICC) showed a very good to excellent agreement observed between all the four readers, for the BI-RADS final assessment category, with a slightly higher agreement observed when using AI software. Without AI software, the average measures ICC was 0.910 with a 95% CI of 0.894 to 0.924, F(433, 1299), p < 0.001. With AI software, the average measures ICC was 0.931 with a 95% CI of 0.920 to 0.941, F (433, 1299), p < 0.001. The significant p-values from the ANOVA tests indicate that the differences in measurements between the readers with and without AI software are statistically significant. Table 3 presents the frequency distribution of BI-RADS categories assigned by each reader, both with and without AI assistance. Fig 2 provides a visual representation of BI-RADS category distribution, facilitating comparison between AI-assisted and non-AI interpretations.

## Diagnostic performance overall

AI assistance improved the diagnostic performance of all reader groups (Table 4, Fig 3). This is evident from higher sensitivity, specificity, PPV, and AUC for all reader groups with AI (R1AI, R2AI, R3AI, R4AI) compared to without AI (R1, R2, R3, R4). General radiologist (R2), with 12 years of experience, achieved the highest sensitivity and specificity, both with and without AI assistance, suggesting R2 was the most accurate in distinguishing between malignant and benign cases. General radiologist (R3), with 6 years of experience, had the highest PPV and NPV, indicating better accuracy in confirming true positive and true negative cases, both with and without AI assistance. Trainee radiologists (R1 and R4) exhibited the lowest performance, both with and without AI assistance, likely due to less experience. However, AI significantly improved their sensitivity and PPV. Fig 3 (radar plot), provides a visual comparison of overall diagnostic performance, illustrating the increase in key metrics with AI assistance. An example of AI-assisted detection in a patient with a fatty breast composition is illustrated in Fig 4. The AI system successfully identified high-density lesions, with findings confirmed by radiologists both with and without AI assistance.

**Table 3. Distribution of final BI-RADS assessment categories assigned by four readers (R1, R2, R3, R4) with and without AI assistance. The table presents the frequency of BI-RADS classifications for each reader, illustrating differences in interpretation when using AI.**

| BI-RADS Category | READER STUDY WITHOUT AI (R1, R2, R3, R4) and WITH AI (R1AI, R2A1, R3A1, R4AI) (Frequency n (%)) | | | | | | | |
|---|---|---|---|---|---|---|---|---|
| | R1 | R1 AI | R2 | R2 AI | R3 | R3 AI | R4 | R4AI |
| 1 | 115 (26.5) | 106 (24.4) | 114 (26.3) | 112 (25.8) | 90 (20.7) | 123 (28.3) | 24 (5.5) | 56 (12.9) |
| 2 | 214 (49.3) | 174 (40.1) | 188 (43.3) | 218 (50.2) | 242 (55.8) | 240 (55.5) | 270 (62.2) | 249 (57.4) |
| 3 | 10 (2.3) | 42 (9.7) | 36 (8.3) | 16 (3.7) | 28 (6.5) | 8 (1.8) | 73 (16.8) | 58 (13.4) |
| 4a | 20 (4.6) | 25 (5.8) | 15 (3.5) | 20 (4.6) | 8 (1.8) | 7 (1.6) | 53 (12.2) | 39 (9.0) |
| 4b | 37 (8.5) | 23 (5.3) | 11 (2.5) | 27 (6.2) | 16 (3.7) | 13 (3.0) | 7 (1.6) | 19 (4.4) |
| 4c | 2 (0.5) | 38 (8.8) | 44 (10.1) | 1 (0.2) | 28 (6.5) | 14 (3.2) | 5 (1.2) | 8 (1.8) |
| 5 | 36 (8.3) | 24 (5.5) | 24 (5.5) | 40 (9.2) | 20 (4.6) | 29 (6.7) | 2 (0.5) | 5 (1.2) |

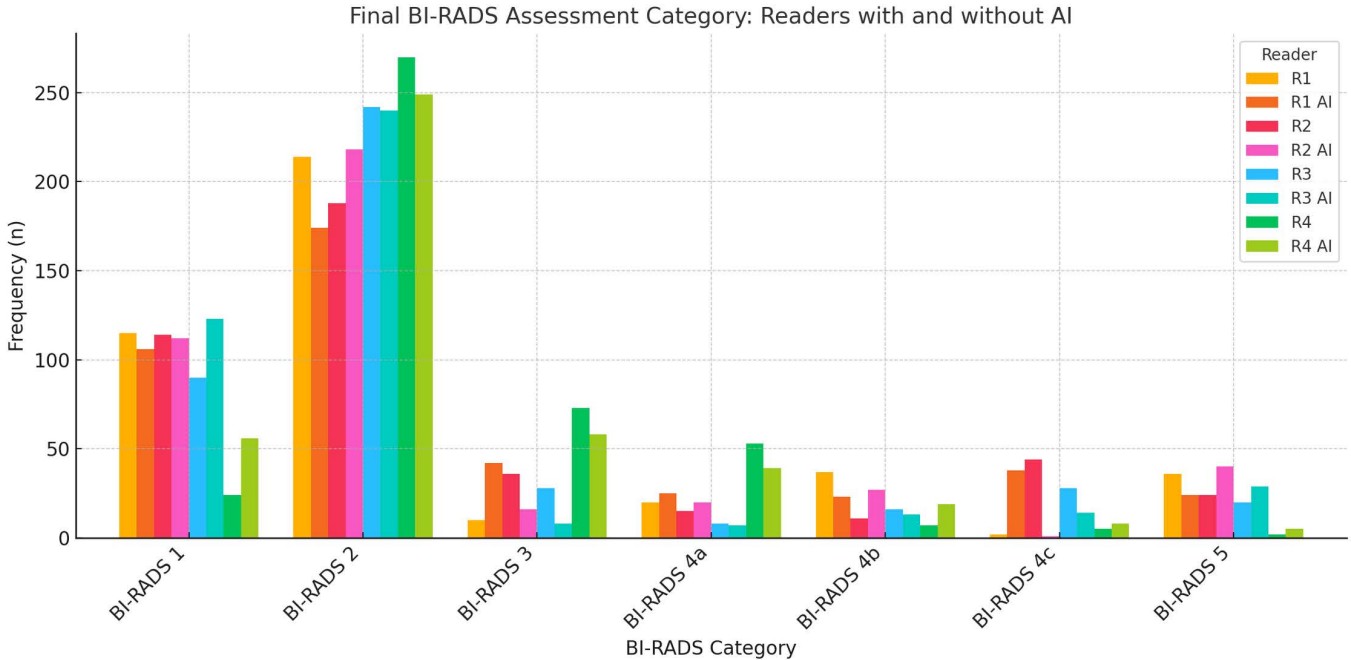

**Fig 2. Bar chart showing the distribution of final BI-RADS assessment categories among radiologists (R1, R2, R3, R4) with and without AI assistance.** The graph highlights differences in classification patterns, illustrating trends in BI-RADS assignments based on AI involvement.

## Diagnostic performance in dense breast

AI assistance enhanced performance across all reader groups for mammograms of women with dense breasts (Table 5, Fig 5). General radiologist (R2) continued to show the highest sensitivity and specificity, indicating superior accuracy in classifying malignant and benign cases even in dense breasts. General radiologist (R3) showed higher PPV without AI assistance, but higher NPV with AI, suggesting AI's role in reducing false negatives in dense breast assessments. Trainee radiologists (R1 and R4) performed the lowest overall, but AI improved their diagnostic accuracy, particularly in terms of NPV and AUC. The radar plot (Fig 5) visually compares the diagnostic performance in dense breast, demonstrating how AI assistance enhances detection but with varying impacts depending on reader experience. Fig 6 demonstrates

**Table 4. Overall diagnostic performance metrics of four radiologists (R1, R2, R3, R4) with and without AI assistance. The table presents sensitivity, specificity, PPV, NPV, and AUC values, illustrating the impact of AI across different readers.**

| | READERS DIAGNOSTIC PERFORMANCE WITHOUT AI (R1, R2, R3, R4) and WITH AI (R1AI, R2A1, R3A1, R4AI) (%) | | | | | | | | | | |
|---|---|---|---|---|---|---|---|---|---|---|---|
| | R1 | R1AI | R2 | R2AI | R3 | R3AI | R4 | R4AI | ALL* | ALLAI* | † |
| **Sensitivity** | 87.2 | 84.0 | 86.5 | 88.0 | 91.3 | 95.5 | 91.9 | 92.5 | 89.2 | 90.0 | 0.140 |
| **Specificity** | 64.5 | 69.7 | 60.5 | 59.2 | 53.9 | 61.8 | 50.0 | 57.9 | 57.2 | 62.2 | 0.195 |
| **PPV** | 51.6 | 48.2 | 48.9 | 51.1 | 56.9 | 74.6 | 56.7 | 62.0 | 53.5 | 59.0 | 0.144 |
| **NPV** | 92.0 | 92.9 | 91.1 | 91.0 | 90.3 | 92.2 | 89.6 | 91.2 | 90.8 | 91.9 | 0.313 |
| **AUC** | 0.811 | 0.799 | 0.792 | 0.790 | 0.779 | 0.819 | 0.741 | 0.814 | 0.780 | 0.806 | 0.285 |

*All indicates the mean of all readers

AI-assisted interpretation in a patient with a dense breast composition. AI highlighted multiple suspicious regions, supporting radiologists in categorizing this case as suspicious, consistent with non-AI assessments.

The sensitivity and specificity values were lower for all reader groups in women with dense breasts compared to overall performance, suggesting greater challenges in interpreting mammograms in this subgroup. AI assistance led to an improvement in diagnostic performance across all metrics, though the extent of improvement varied among readers.

The trend of AI-assisted improvement is clearly visualized in the radar plots (Figs 3 and 5), where AI-assisted interpretation resulted in higher sensitivity, specificity, and AUC across both overall and dense breast subgroups. These figures illustrate the differences in AI's impact on performance across different breast densities and reader experience levels.

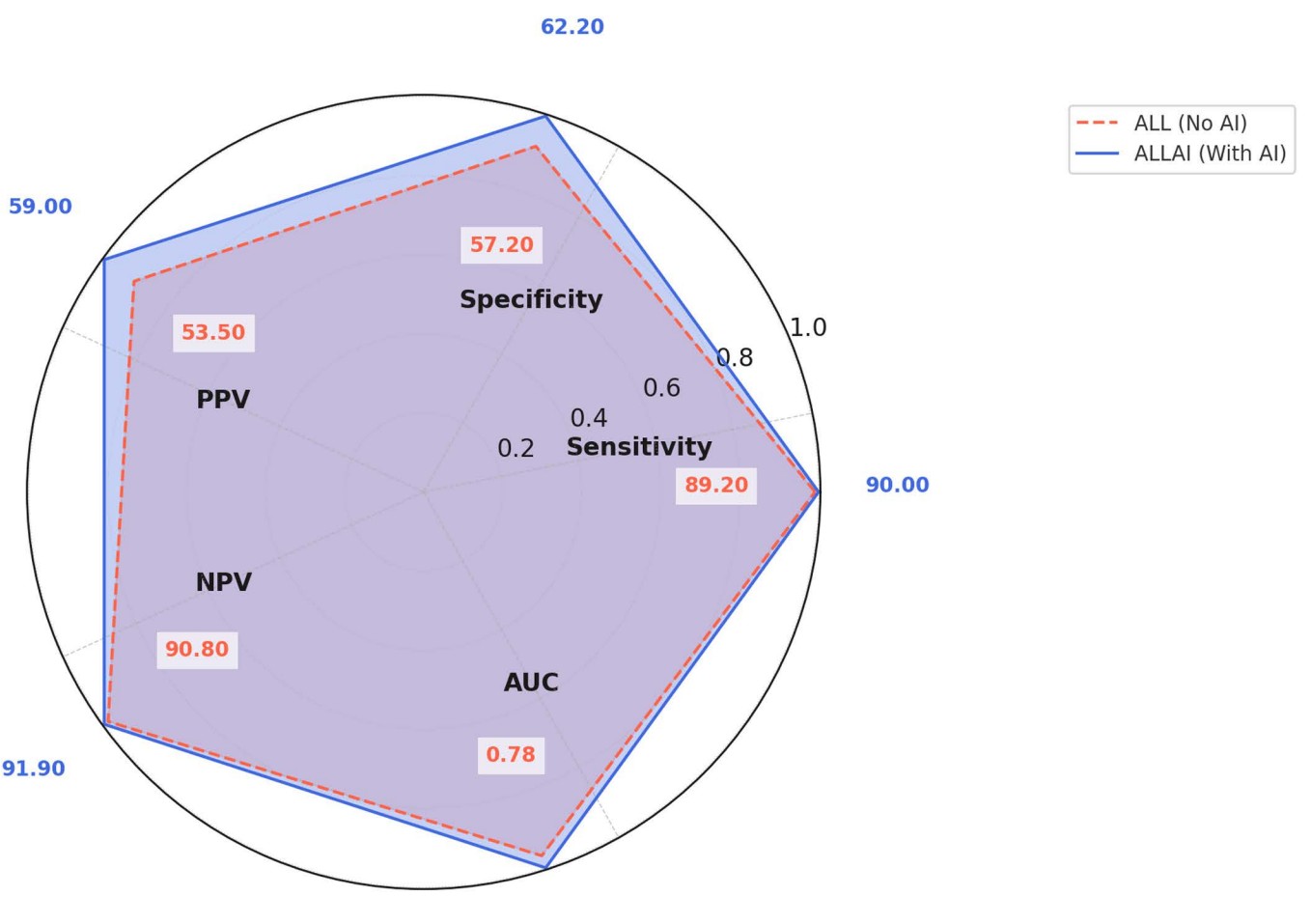

**Fig 3. Radar plot comparing overall diagnostic performance with and without AI assistance for all readers.** The graph illustrates key performance metrics, including Sensitivity, Specificity, PPV, NPV, and AUC. The blue line represents AI-assisted performance, while the red dashed line represents performance without AI assistance.

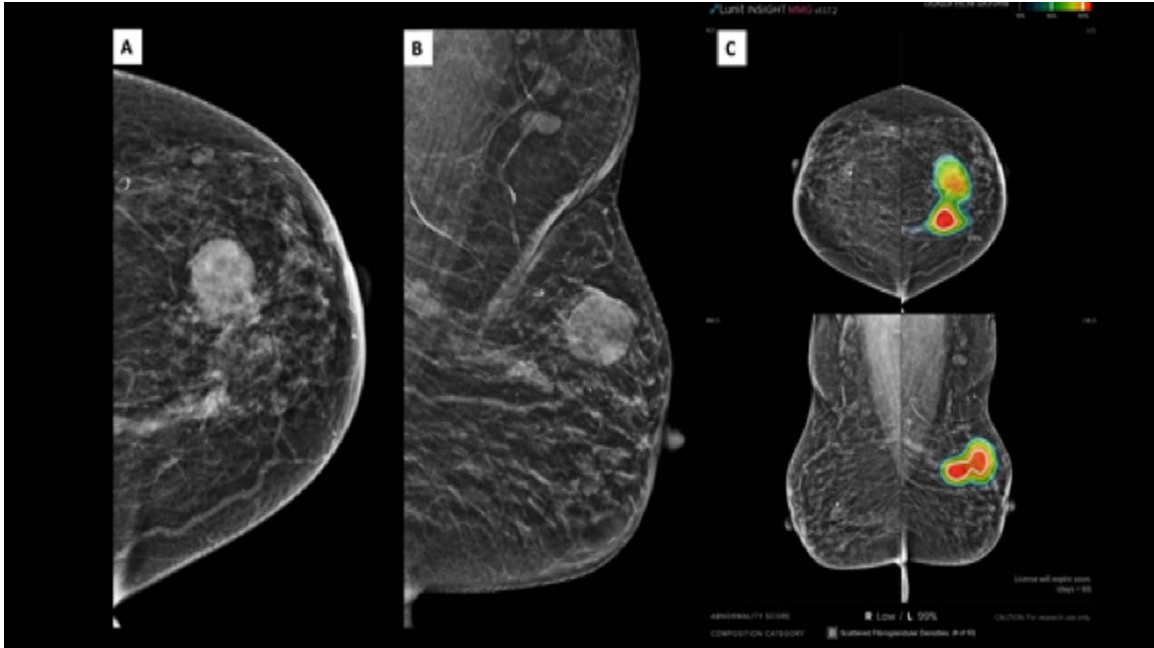

**Fig 4. Mammogram of a patient with a fatty breast composition.** Craniocaudal (A) and mediolateral oblique (B) views show two high-density lesions in the left breast. AI-assisted analysis (C) highlights the suspicious regions with a high abnormality score, prompting further evaluation. Both AI-assisted and non-AI interpretations categorized this case as suspicious.

## Discussion

AI has demonstrated significant potential in improving breast cancer screening, particularly by aiding less experienced radiologists [2,17–20]. However, its effect varies based on radiologist expertise, dataset quality, and mammographic findings [3,4,6]. Automation bias and the potential over-reliance on AI are critical considerations in AI-assisted radiology. While AI can enhance diagnostic accuracy, excessive dependence on AI outputs may reduce radiologists' critical thinking and decision-making skills, potentially impacting patient care [19]. The observed slight decrease in AUC for some readers suggests that AI may not always enhance performance and could introduce unintended challenges in interpretation [19].

One critical factor influencing AI's performance in mammography interpretation is image quality. High-quality images with good contrast resolution enable AI to detect subtle abnormalities more effectively, whereas poor-quality

**Table 5. Diagnostic performance metrics in dense breasts, comparing four radiologists (R1, R2, R3, R4) with and without AI assistance. The table presents sensitivity, specificity, PPV, NPV, and AUC, illustrating AI's role in evaluating mammograms in dense breast tissue.**

|  | READERS DIAGNOSTIC PERFORMANCE IN DENSE BREAST WITHOUT AI (R1, R2, R3, R4) and WITH AI (R1AI, R2A1, R3A1, R4AI) (%) | | | | | | | | | | |
|---|---|---|---|---|---|---|---|---|---|---|---|
|  | R1 | R1AI | R2 | R2AI | R3 | R3AI | R4 | R4AI | ALL* | AI* | † |
| **Sensitivity** | 83.8 | 80.3 | 84.6 | 85.1 | 89.3 | 94.0 | 90.6 | 91.5 | 87.1 | 87.7 | 0.213 |
| **Specificity** | 56.6 | 66.0 | 56.6 | 49.1 | 50.9 | 58.5 | 45.3 | 54.7 | 52.4 | 57.1 | 0.238 |
| **PPV** | 45.5 | 43.2 | 51.9 | 42.6 | 52.2 | 68.9 | 43.2 | 59.2 | 48.2 | 53.5 | 0.213 |
| **NPV** | 89.6 | 88.1 | 88.9 | 88.1 | 88.0 | 90.9 | 91.3 | 90.0 | 89.5 | 89.3 | 0.213 |
| **AUC** | 0.754 | 0.764 | 0.781 | 0.739 | 0.738 | 0.789 | 0.700 | 0.796 | 0.743 | 0.772 | 0.213 |

*All indicates the mean of all readers

images—especially in cases of dense breast tissue—can reduce AI's diagnostic accuracy [5,7]. Mammographic noise, artifacts, and varying exposure levels may contribute to false-positive or false-negative results [8]. In particular, dense breast tissue can obscure lesions, making it difficult for both AI and human readers to achieve high sensitivity [9,10]. The performance of AI systems should, therefore, be continuously evaluated across different imaging conditions to ensure optimal reliability [17].

Furthermore, the variability in AI impact across different readers highlights the importance of experience level when integrating AI into radiology workflows [3,6]. While AI has been particularly beneficial for trainee radiologists by improving their sensitivity and overall performance, its effect on experienced radiologists has been more variable [4,18]. This suggests that AI should serve as an adjunct rather than a replacement for human interpretation, ensuring that radiologists remain actively engaged in decision-making rather than becoming over-reliant on AI outputs [19]. This variability in AI

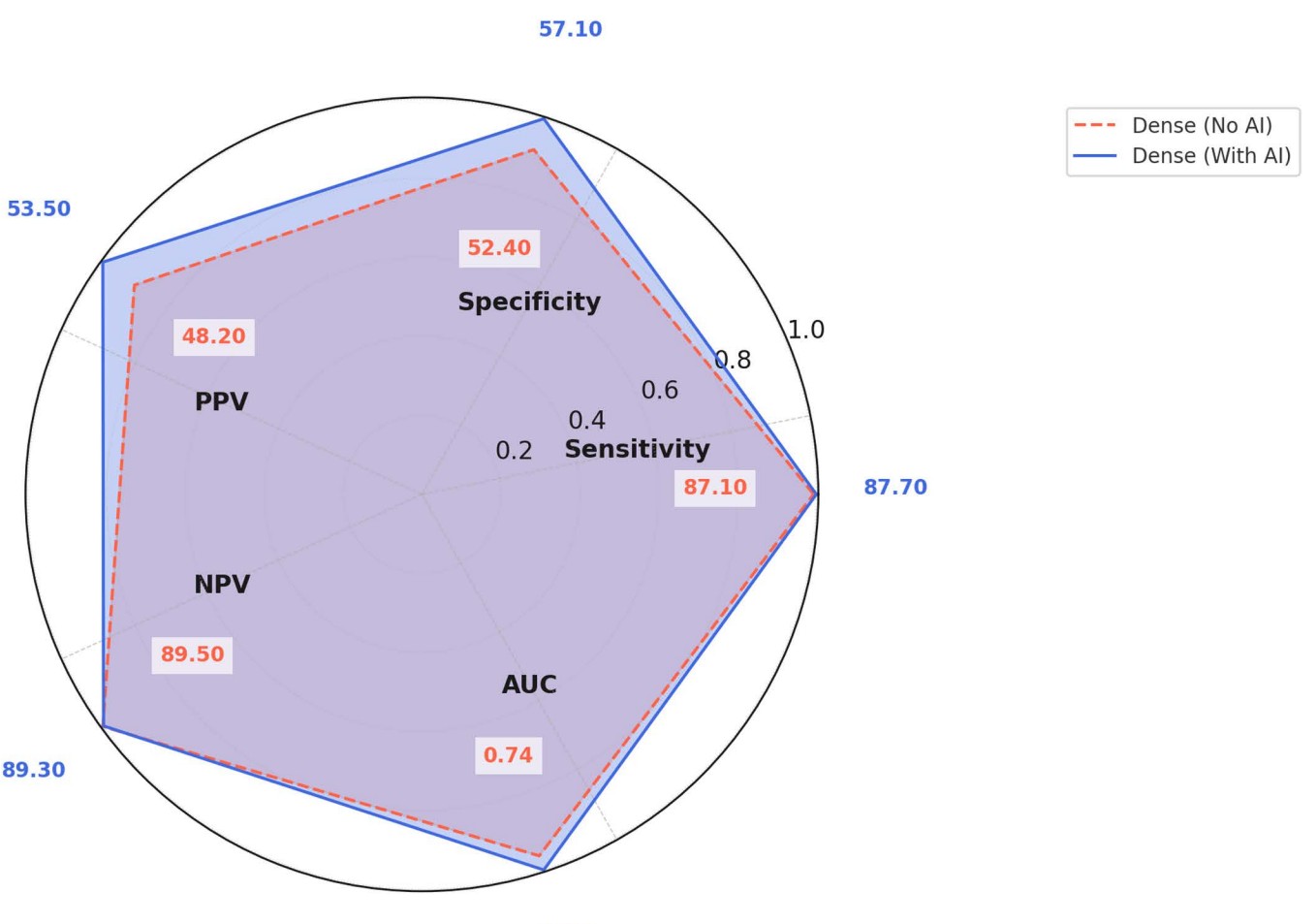

**Fig 5. Radar plot comparing diagnostic performance in dense breasts for all readers (R1, R2, R3, R4) with and without AI assistance.** The figure illustrates how AI affects Sensitivity, Specificity, PPV, NPV, and AUC when interpreting dense breast cases. The blue line represents AI-assisted performance, while the red dashed line represents performance without AI assistance.

performance may explain the observed decrease in AUC values for R1 and R2, as shown in Tables 4 and 5, where AI's impact was not uniformly positive.

The study findings also emphasize the potential role of AI in resource-limited settings, such as Malaysia and other low- and middle-income countries with a shortage of dedicated breast radiologists [1,7,9]. In such environments, AI can function as a second reader, helping to bridge the gap in expertise and reducing diagnostic discrepancies between general radiologists and breast imaging specialists [20]. The insights gained from this study may be generalizable to other countries with similar healthcare challenges, reinforcing AI's role in addressing global disparities in breast cancer detection [10,17].

The performance variations observed among different readers in our study also highlight the potential limitations of AI assistance. Some radiologists, particularly those with more experience, may interpret AI outputs differently or be more cautious in adjusting their diagnoses. Additionally, the observed decrease in AUC for some readers suggests that AI's effectiveness depends on user interaction and experience level. This underscores the need for further training and calibration of AI tools to optimize their integration into clinical practice.

### Limitations and future research directions

This study has several limitations. As a single-center retrospective study, the findings may not be fully generalizable to broader populations. Additionally, the lack of direct comparison with other AI models limits the ability to conclude whether Lunit INSIGHT MMG performs better than alternative AI systems. Another concern is the potential for automation bias, particularly if radiologists become overly reliant on AI predictions rather than using them as a supportive tool. Finally, there is a need for prospective, multicenter validation studies to assess AI's clinical impact across diverse patient populations and real-world screening settings. Future research should explore AI's role in multicenter trials, evaluate its

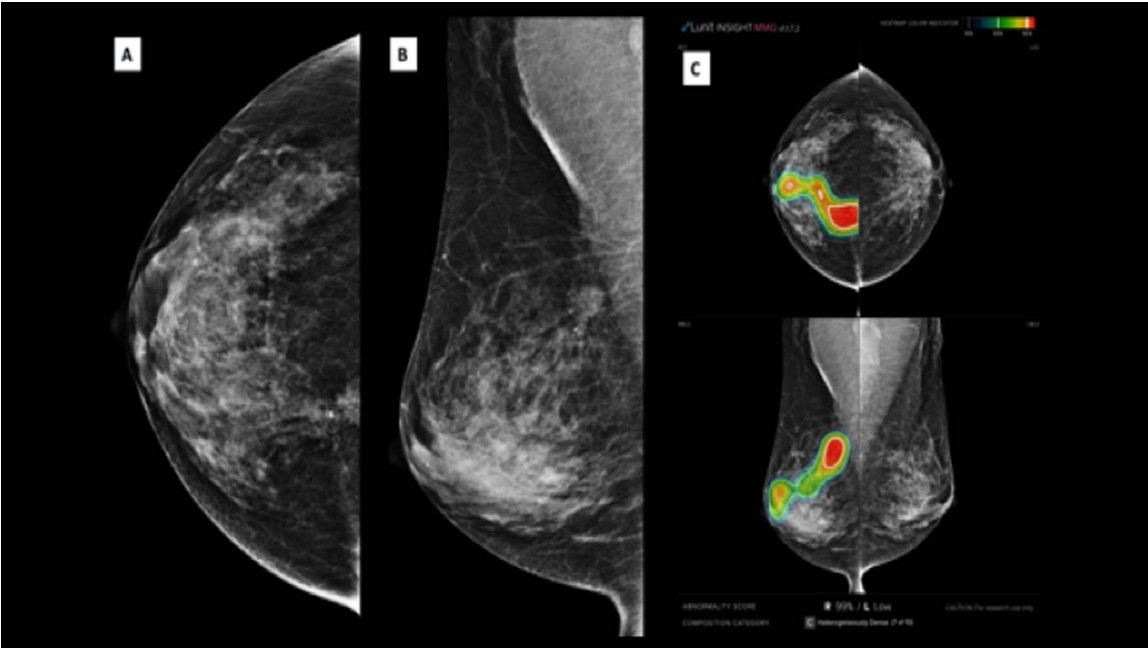

**Fig 6. Mammogram of a patient with a dense breast composition.** Craniocaudal (A) and mediolateral oblique (B) views reveal multiple high-density lesions in the right breast. AI-assisted analysis (C) assigns a high abnormality score, supporting the identification of suspicious areas. Both AI-assisted and non-AI interpretations categorized this case as suspicious.

cost-effectiveness, and examine its integration into routine radiological workflows to determine its long-term viability in breast cancer screening programs.

## Conclusion

his study highlights the significant role of artificial intelligence in enhancing breast cancer screening, particularly in resource-limited settings like Malaysia. AI has demonstrated the potential to improve diagnostic accuracy, reduce inter-reader variability, and support less experienced radiologists, contributing to a more standardized and equitable healthcare system.

Beyond Malaysia, these findings are relevant to other low- and middle-income countries facing similar challenges. AI can help bridge the expertise gap and optimize resources, facilitating earlier cancer detection and improved patient outcomes. However, careful implementation is necessary to prevent automation bias and ensure AI complements rather than replaces human expertise.

While this study provides valuable insights, further research, including multi-center trials and cost-effectiveness studies, is needed to establish AI's long-term clinical impact. Strategic adoption and ongoing refinement will be crucial to maximizing AI's potential in breast cancer detection and patient care.

## Author contributions

**Conceptualization:** Marlina Tanty Ramli Hamid, Nazimah AB Mumin.

**Data curation:** Marlina Tanty Ramli Hamid, Nazimah AB Mumin.

**Formal analysis:** Marlina Tanty Ramli Hamid, Nazimah AB Mumin, Shamsiah Abdul Hamid.

**Funding acquisition:** Marlina Tanty Ramli Hamid, Nazimah AB Mumin, Shamsiah Abdul Hamid.

**Investigation:** Marlina Tanty Ramli Hamid, Nazimah AB Mumin, Shamsiah Abdul Hamid, Natasha Mohd Ariffin, Khariah Mat Nor, Ernisha Saib, Nurul Amira Mohamed.

**Methodology:** Marlina Tanty Ramli Hamid, Nazimah AB Mumin, Shamsiah Abdul Hamid, Natasha Mohd Ariffin, Khariah Mat Nor, Ernisha Saib, Nurul Amira Mohamed.

**Project administration:** Marlina Tanty Ramli Hamid, Nazimah AB Mumin.

**Resources:** Marlina Tanty Ramli Hamid, Nazimah AB Mumin.

**Supervision:** Marlina Tanty Ramli Hamid, Nazimah AB Mumin.

**Writing – original draft:** MARLINA TANTY RAMLI HAMID.

**Writing – review & editing:** Nazimah AB Mumin, Shamsiah Abdul Hamid, Natasha Mohd Ariffin, Khariah Mat Nor, Ernisha Saib, Nurul Amira Mohamed.

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
