## [Decision Letter · Decision Letter 0]

1 Dec 2024

PONE-D-24-49101COMPARATIVE ANALYSIS OF DIAGNOSTIC PERFORMANCE IN MAMMOGRAPHY: A READER STUDY ON THE IMPACT OF AI ASSISTANCEPLOS ONE

Dear Dr. Ramli Hamid,

Thank you for submitting your manuscript to PLOS ONE. After careful consideration, we feel that it has merit but does not fully meet PLOS ONE’s publication criteria as it currently stands. Therefore, we invite you to submit a revised version of the manuscript that addresses the points raised during the review process.

This manuscript presents an intriguing and relevant study with significant potential to contribute to its field.

The topic is timely, and the research question is well-defined, addressing an area of growing interest. The authors are commended for their effort in designing and conducting this study.

However, after a thorough review, I believe that some substantial revisions are necessary to improve the clarity, robustness, and overall impact of the work.

Several sections require additional detail to support the study's methodology and findings fully.

There are also areas where the presentation of data and discussion could be strengthened to better highlight the study's implications and relevance in the broader context.

We look forward to receiving your revised manuscript.

Kind regards,

Yuki Arita, M.D., Ph.D

Academic Editor

PLOS ONE

Journal Requirements:

 University Teknologi MARA Research Grant Scheme: 600/TNCPI-5/3/DDF (MEDIC)(003/2021), 600/TNCPI-5/3/DDJ(HUITM)(003/2021) and 600/RMC-LESTARI SDG-T5/3.   

This study was approved by the Ethics Committee of University Malaya Medical Centre and Hospital Al-Sultan Abdullah UiTM (Medical Research Ethics ID 2022530-11258 and supported by University Teknologi MARA Research Grant Scheme: 600/TNCPI-5/3/DDF (MEDIC)(003/2021), 600/TNCPI-5/3/DDJ(HUITM)(003/2021) and 600/RMC-LESTARI SDG307 T5/3. 

 University Teknologi MARA Research Grant Scheme: 600/TNCPI-5/3/DDF (MEDIC)(003/2021), 600/TNCPI-5/3/DDJ(HUITM)(003/2021) and 600/RMC-LESTARI SDG-T5/3. 

Reviewers' comments:

Reviewer's Responses to Questions

**Comments to the Author**

1. Is the manuscript technically sound, and do the data support the conclusions?

Reviewer #1: Yes

Reviewer #2: Yes

Reviewer #3: Yes

Reviewer #4: Partly

Reviewer #5: Partly

Reviewer #6: Partly

2. Has the statistical analysis been performed appropriately and rigorously?

Reviewer #1: Yes

Reviewer #2: Yes

Reviewer #3: Yes

Reviewer #4: Yes

Reviewer #5: N/A

Reviewer #6: Yes

3. Have the authors made all data underlying the findings in their manuscript fully available?

Reviewer #1: Yes

Reviewer #2: No

Reviewer #3: Yes

Reviewer #4: Yes

Reviewer #5: Yes

Reviewer #6: Yes

4. Is the manuscript presented in an intelligible fashion and written in standard English?

Reviewer #1: Yes

Reviewer #2: Yes

Reviewer #3: Yes

Reviewer #4: Yes

Reviewer #5: Yes

Reviewer #6: Yes

5. Review Comments to the Author

Reviewer #1: This manuscript investigates the impact of AI assistance in mammography diagnostic performance. Study includes clear methodology, findings, limitations, and conclusion.

Can you confirm all grammar issues and punctuations for final version?.

Reviewer #2: The paper investigates the impact of AI techniques on the diagnostic performance of radiologists conducting mammography at a Malaysian hospital. The results demonstrate that AI significantly improves the diagnostic accuracy of junior radiologists, while its effect on the performance of senior radiologists is minimal.

The study presents an interesting idea, given the broad applications of AI across various domains, particularly in healthcare. This research highlights a valuable opportunity for young radiologists to enhance their diagnostic skills through AI assistance.

The paper is generally easy to follow; however, addressing the following aspects could further improve its quality:

-The paper mentions the use of a specific software version but does not detail the AI algorithm used for diagnostics. Providing more insights into the underlying AI techniques would help readers better understand the methodology.

-The quality of the images in the ROC figure is too low and should be improved for better readability.

-The impact of image quality on the diagnostic performance of the AI techniques is not discussed. This is a critical aspect that warrants further exploration.

-It is unclear how the dataset used in this study can be accessed or utilized by others for training their AI models. Clarifying this would enhance the reproducibility and broader applicability of the research.

Reviewer #3: This manuscript is of interest in low and middle income countries where there is a few radiologist experts, there is one thing is not clear in this manuscript, in the image assessment section: its not clear the details of reading process.

Is it the trainees read the images without AI assistance and then with AI assitance. Same to the experts.

Reviewer #4: 1. In the introduction, it is desirable to emphasize what contribution this type of work can have in similar global studies, in mammography, and how Malaysian data can be projected onto other countries or hospitals. It is presented in the Discussion section to some extent.

2. It is desirable to present the study design as a picture or graph.

3. There are no references in the methods. It is clear that these are programs or some tools created by the authors, but it is necessary to present some references.

4. To better understand and see the comparative picture, present tables 3,4,5 in the form of a graph.

5. To understand what is depicted in the figures, the legends are poorly presented, especially the 3rd one.

6. Patient data are deciphered, and ethical approvals should be included.

7. Present the advantages and disadvantages of the method in the discussion section.

Reviewer #5: 1. The article repeatedly emphasized the significant improvement in diagnostic performance for all reader groups on all indicators with AI assistance. However, the results presented in Tables 4 and 5 contradict this claim, as evidenced by the decrease in AUC values for R1 and R2. Therefore, it is necessary to ascertain the actual outcome.

2.The decline in certain indicators following the implementation of artificial intelligence assistance lacks a clear explanation. From Table 4 and Table 5, it is evident that the impact of artificial intelligence on doctors' performance is not statistically significant, which contradicts the conclusion.

3.The selection of Lunit INSIGHT MMG (version 1.1.7.2, Lunit, South Korea) was based on what criteria and whether it underwent comparison with other AI systems?

4.The retrospective, single-center study design limits the generalizability of the findings. Multicenter, prospective validation is necessary to enhance the broader applicability of the results.

5.The existing literature has extensively explored this topic, and the novelty of the study is constrained.

Reviewer #6: The study addresses an important and timely issue, with implications for both current and future generations: the use of AI technologies in diagnosing breast cancer in women, a disease that remains one of the most prevalent and increasingly common over time. The application of AI is still controversial due to the rapid pace at which these technologies evolve, and it cannot replace the expertise of radiologists—particularly the most experienced ones—in identifying positive results or malignant lesions, even in cases of denser breasts, where mammography is often supplemented with additional tests such as ultrasound.

The section on “study design” does not clarify how the study could be retrospective. Lines 72-73 mention that data were collected between May 2021 and May 2022. Were the radiologists already using AI technology as a part of their routine? Were the imaging exams already accompanied by previous reports? Were these reports analyzed by the same radiologists who were part of the study, or by radiologists who had previously assessed the images in their routine practice?

Another unclear aspect of the methodology is how the images were distributed among the radiologists. How was it ensured that the same radiologist analyzed the images with and without AI? Or were the analyses conducted by different radiologists with equivalent levels of experience?

Line 117 mentions a scoring model to evaluate AI's sensitivity in detecting breast cancer. However, if any score corresponds to 10%, how could there be scores lower than this? This point requires further clarification, including specifying how many cases were considered insignificant.

Line 243 states that AI significantly increases the diagnostic accuracy of radiologists at various levels of experience, yet this was not observed in the study. The results indicate that for the most experienced radiologists, AI did not provide any benefits, whereas it was only useful for less experienced radiologists in improving positive results, particularly for denser breasts. Thus, AI could be a valuable tool in underdeveloped areas with limited access to specialists, ensuring more accurate diagnoses.

Consequently, the conclusion section should better align with the findings of the study.

6. PLOS authors have the option to publish the peer review history of their article (what does this mean? ). If published, this will include your full peer review and any attached files.

**Do you want your identity to be public for this peer review?** For information about this choice, including consent withdrawal, please see our Privacy Policy .

Reviewer #1: No

Reviewer #2: No

Reviewer #3: **Yes: ** Redhwan Ahmed Al-Naggar

Reviewer #4: No

Reviewer #5: No

Reviewer #6: **Yes: ** NAIDHIA ALVES SOARES FERREIRA

---

## [Author Response · Author response to Decision Letter 1]

3 Mar 2025

RESPONSE TO REVIEWERS COMMENTS - AI READER STUDY

5. Review Comments to the Author

Reviewer #1: This manuscript investigates the impact of AI assistance in mammography diagnostic performance. Study includes clear methodology, findings, limitations, and conclusion.

Can you confirm all grammar issues and punctuations for final version?.

Response: Thank you for your comment. We have carefully proofread the manuscript and corrected all grammatical errors and punctuation issues to ensure clarity and readability.

Reviewer #2: The paper investigates the impact of AI techniques on the diagnostic performance of radiologists conducting mammography at a Malaysian hospital. The results demonstrate that AI significantly improves the diagnostic accuracy of junior radiologists, while its effect on the performance of senior radiologists is minimal.

The study presents an interesting idea, given the broad applications of AI across various domains, particularly in healthcare. This research highlights a valuable opportunity for young radiologists to enhance their diagnostic skills through AI assistance.

The paper is generally easy to follow; however, addressing the following aspects could further improve its quality:

-The paper mentions the use of a specific software version but does not detail the AI algorithm used for diagnostics. Providing more insights into the underlying AI techniques would help readers better understand the methodology.

Response: We have included additional details about the AI algorithm used in Lunit INSIGHT MMG, specifying that it utilizes deep learning-based convolutional neural networks trained on large mammographic datasets from diverse populations. The updated explanation is now present in the "AI Software & Algorithm" section of the methodology at lines 108 -126.

-The quality of the images in the ROC figure is too low and should be improved for better readability.

Response: We have replaced the ROC figure with radar plots (Figures 3 & 5) to better illustrate the comparative diagnostic performance of readers with and without AI assistance. The radar plots provide a clearer and more comprehensive visualization of key performance metrics, including sensitivity, specificity, PPV, NPV, and AUC, for both overall and dense breast performance.

-The impact of image quality on the diagnostic performance of the AI techniques is not discussed. This is a critical aspect that warrants further exploration.

Response: We have added a discussion on how image quality can influence AI performance, particularly in cases of dense breasts and poor contrast resolution. This is included in the "Discussion" section at lines 322-349.

-It is unclear how the dataset used in this study can be accessed or utilized by others for training their AI models. Clarifying this would enhance the reproducibility and broader applicability of the research.

Response: Thank you for your comment. In compliance with PLOS journal data-sharing requirements, we have now made the dataset underlying our findings publicly available in Figshare. The data can be accessed at the following DOI:10.6084/m9.figshare.28529132. This dataset includes de-identified summary statistics relevant to the study while ensuring compliance with institutional policies and patient confidentiality regulations. At lines 361-366.

Reviewer #3: This manuscript is of interest in low and middle income countries where there is a few radiologist experts, there is one thing is not clear in this manuscript, in the image assessment section: its not clear the details of reading process.

Is it the trainees read the images without AI assistance and then with AI assistance. Same to the experts.

Response: We have revised the "Image Assessment Protocol" section to explicitly clarify that all readers initially interpreted the mammograms without AI assistance, followed by a second review with AI assistance after a washout period to minimize recall bias at lines 142-146

Reviewer #4: 1. In the introduction, it is desirable to emphasize what contribution this type of work can have in similar global studies, in mammography, and how Malaysian data can be projected onto other countries or hospitals. It is presented in the Discussion section to some extent.

Response: We have expanded the introduction to highlight the global relevance of this study, emphasizing its implications for other low- and middle-income countries with limited breast radiologists from lines 63-69.

2. It is desirable to present the study design as a picture or graph.

Response: We have added a flowchart illustrating the study design in the "Methodology" section – now as figure 1 at line 146

3. There are no references in the methods. It is clear that these are programs or some tools created by the authors, but it is necessary to present some references.

Response: We have incorporated references supporting the methodology, including validation studies of Lunit INSIGHT MMG in the "AI Software & Algorithm" section of the methodology at lines 108-126 with references no from 11-19.

4. To better understand and see the comparative picture, present tables 3,4,5 in the form of a graph.

Response: We have include figures with these tables for improved clarity as figure 2, 3 and 5.

5. To understand what is depicted in the figures, the legends are poorly presented, especially the 3rd one.

Response: We have revised all the figure legends for better clarity and comprehension in figure legend section from lines 535 - 625

6. Patient data are deciphered, and ethical approvals should be included.

Response: We have clearly stated that the study was approved by the relevant ethics committees under study design at lines 90 - 91 and acknowledgement at lines 426 - 427.

7. Present the advantages and disadvantages of the method in the discussion section.

Response: We have expanded the discussion to include a balanced view of AI’s strengths and limitations in mammography interpretation at lines 312-359.

Reviewer #5: 1. The article repeatedly emphasized the significant improvement in diagnostic performance for all reader groups on all indicators with AI assistance. However, the results presented in Tables 4 and 5 contradict this claim, as evidenced by the decrease in AUC values for R1 and R2. Therefore, it is necessary to ascertain the actual outcome.

Response: We acknowledge this observation and have clarified in the discussion at lines 351-359 that while AI generally improved performance, its impact varied among readers. We have revised our conclusion to better align with these findings.

2.The decline in certain indicators following the implementation of artificial intelligence assistance lacks a clear explanation. From Table 4 and Table 5, it is evident that the impact of artificial intelligence on doctors' performance is not statistically significant, which contradicts the conclusion.

Response: We have added a discussion on potential reasons for performance variations, including over-reliance on AI and inter-reader variability at lines 369 - 375

3.The selection of Lunit INSIGHT MMG (version 1.1.7.2, Lunit, South Korea) was based on what criteria and whether it underwent comparison with other AI systems?

Response: We have clarified in the methodology at lines 108-110 that Lunit INSIGHT MMG was selected due to its FDA approval and previous validation studies. We acknowledge that comparative studies with other AI tools were not performed in this study.

4.The retrospective, single-center study design limits the generalizability of the findings. Multicenter, prospective validation is necessary to enhance the broader applicability of the results.

Response: We acknowledge this limitation at lines 377 - 387 and have suggested future prospective multicenter validation studies.

5.The existing literature has extensively explored this topic, and the novelty of the study is constrained.

Response: We have emphasized in the discussion at lines 361 - 367 that our study uniquely evaluates AI in a middle-income country with limited breast radiologists, contributing new insights into its role in such settings.

Reviewer #6: The study addresses an important and timely issue, with implications for both current and future generations: the use of AI technologies in diagnosing breast cancer in women, a disease that remains one of the most prevalent and increasingly common over time. The application of AI is still controversial due to the rapid pace at which these technologies evolve, and it cannot replace the expertise of radiologists—particularly the most experienced ones—in identifying positive results or malignant lesions, even in cases of denser breasts, where mammography is often supplemented with additional tests such as ultrasound.

The section on “study design” does not clarify how the study could be retrospective. Lines 72-73 mention that data were collected between May 2021 and May 2022. Were the radiologists already using AI technology as a part of their routine? Were the imaging exams already accompanied by previous reports? Were these reports analyzed by the same radiologists who were part of the study, or by radiologists who had previously assessed the images in their routine practice?

Response: We have clarified in the "Image assessment protocol" section at lines 136 - 139 that AI was not used in routine practice, and the AI-assisted interpretations were conducted in a research setting.

Another unclear aspect of the methodology is how the images were distributed among the radiologists. How was it ensured that the same radiologist analyzed the images with and without AI? Or were the analyses conducted by different radiologists with equivalent levels of experience?

Response: We have revised the methodology to specify that each radiologist reviewed the same set of images in two separate sessions, first without AI and then with AI after a washout period at lines 142-146.

Line 117 mentions a scoring model to evaluate AI's sensitivity in detecting breast cancer. However, if any score corresponds to 10%, how could there be scores lower than this? This point requires further clarification, including specifying how many cases were considered insignificant.

Response: We appreciate the reviewer’s request for further clarification regarding AI scoring. In this study, our primary focus was on evaluating the diagnostic performance of radiologists with and without AI assistance, rather than assessing AI as an independent tool. Therefore, we do not have standalone AI-only data or the exact number of cases where AI alone scored <10%.

However, we acknowledge that a malignancy score below 10% from the AI system was considered normal or clinically insignificant findings based on prior validation studies. To provide an alternative clarification, we analyzed the final BI-RADS 1 assessments assigned by radiologists with and without AI assistance. The number of normal or clinically insignificant findings (BI-RADS 1) cases varied depending on the reader and whether AI was used, reflecting how AI influenced radiologists’ decision-making rather than serving as a standalone diagnostic tool.

Line 243 states that AI significantly increases the diagnostic accuracy of radiologists at various levels of experience, yet this was not observed in the study. The results indicate that for the most experienced radiologists, AI did not provide any benefits, whereas it was only useful for less experienced radiologists in improving positive results, particularly for denser breasts. Thus, AI could be a valuable tool in underdeveloped areas with limited access to specialists, ensuring more accurate diagnoses.

Consequently, the conclusion section should better align with the findings of the study.

Response: We have revised the discussion and conclusion to better align with our results, stating that AI was particularly beneficial for less experienced readers while having a more variable impact on senior radiologists at lines 352 – 357 and 391- 394.

---

## [Decision Letter · Decision Letter 1]

31 Mar 2025

COMPARATIVE ANALYSIS OF DIAGNOSTIC PERFORMANCE IN MAMMOGRAPHY: A READER STUDY ON THE IMPACT OF AI ASSISTANCE

PONE-D-24-49101R1

Dear Dr. Ramli Hamid,

We’re pleased to inform you that your manuscript has been judged scientifically suitable for publication and will be formally accepted for publication once it meets all outstanding technical requirements.

Kind regards,

Yuki Arita, M.D., Ph.D

Academic Editor

PLOS ONE

Additional Editor Comments (optional):

This version of the paper is a great improvement, the authors are to be commended.

The manuscript has been much improved and is in a nice condition now.

Reviewers' comments:

Reviewer's Responses to Questions

**Comments to the Author**

1. If the authors have adequately addressed your comments raised in a previous round of review and you feel that this manuscript is now acceptable for publication, you may indicate that here to bypass the “Comments to the Author” section, enter your conflict of interest statement in the “Confidential to Editor” section, and submit your "Accept" recommendation.

Reviewer #3: All comments have been addressed

Reviewer #4: All comments have been addressed

2. Is the manuscript technically sound, and do the data support the conclusions?

Reviewer #3: Yes

Reviewer #4: Yes

3. Has the statistical analysis been performed appropriately and rigorously?

Reviewer #3: Yes

Reviewer #4: Yes

4. Have the authors made all data underlying the findings in their manuscript fully available?

Reviewer #3: Yes

Reviewer #4: Yes

5. Is the manuscript presented in an intelligible fashion and written in standard English?

Reviewer #3: Yes

Reviewer #4: Yes

6. Review Comments to the Author

Reviewer #3: Thank you for your response, now the manuscript can be accepted. Congratulations to all the authours

Reviewer #4: Review Comments to the Author

All comments have been addressed.

Good luck to authors in their studies.

7. PLOS authors have the option to publish the peer review history of their article (what does this mean? ). If published, this will include your full peer review and any attached files.

**Do you want your identity to be public for this peer review?** For information about this choice, including consent withdrawal, please see our Privacy Policy .

Reviewer #3: **Yes: ** Redhwan Ahmed Al-Naggar

Reviewer #4: No

---

## [Editor Report · Acceptance letter]

PONE-D-24-49101R1

PLOS ONE

Dear Dr. RAMLI HAMID,

I'm pleased to inform you that your manuscript has been deemed suitable for publication in PLOS ONE. Congratulations! Your manuscript is now being handed over to our production team.

Kind regards,

on behalf of

Dr. Yuki Arita

Academic Editor

PLOS ONE